# Neurochemical Basis of Inter-Organ Crosstalk in Health and Obesity: Focus on the Hypothalamus and the Brainstem

**DOI:** 10.3390/cells12131801

**Published:** 2023-07-07

**Authors:** Dhanush Haspula, Zhenzhong Cui

**Affiliations:** 1Molecular Signaling Section, Laboratory of Bioorganic Chemistry, National Institute of Diabetes and Digestive and Kidney Diseases, Bethesda, MD 20892, USA; 2Mouse Metabolism Core, National Institute of Diabetes and Digestive and Kidney Diseases, Bethesda, MD 20892, USA; cuiz@mail.nih.gov

**Keywords:** obesity, hypothalamus, appetite, glucose homeostasis, weight-loss drugs, AGRP, POMC, NTS, incretins

## Abstract

Precise neural regulation is required for maintenance of energy homeostasis. Essential to this are the hypothalamic and brainstem nuclei which are located adjacent and supra-adjacent to the circumventricular organs. They comprise multiple distinct neuronal populations which receive inputs not only from other brain regions, but also from circulating signals such as hormones, nutrients, metabolites and postprandial signals. Hence, they are ideally placed to exert a multi-tier control over metabolism. The neuronal sub-populations present in these key metabolically relevant nuclei regulate various facets of energy balance which includes appetite/satiety control, substrate utilization by peripheral organs and glucose homeostasis. In situations of heightened energy demand or excess, they maintain energy homeostasis by restoring the balance between energy intake and expenditure. While research on the metabolic role of the central nervous system has progressed rapidly, the neural circuitry and molecular mechanisms involved in regulating distinct metabolic functions have only gained traction in the last few decades. The focus of this review is to provide an updated summary of the mechanisms by which the various neuronal subpopulations, mainly located in the hypothalamus and the brainstem, regulate key metabolic functions.

## 1. Introduction

Extensive evidence has unequivocally confirmed the importance of the brain in metabolic disorders and obesity [1]. Research identifying its pathophysiological role has spanned over several decades. From its humble beginnings of employing rodents with hypothalamic lesions, which aided in identifying the role of distinct brain regions in appetite/satiety regulation, to the use of more sophisticated approaches such as chemogenetics and genome-wide association study (GWAS) to identify novel therapeutic targets/pathways in the brain, the central nervous system (CNS) has now been firmly established as a critical component that is dysregulated in the development of obesity [2,3,4]. More recently, research on the metabolic role of the CNS has also paved the way for the identification of drug targets in metabolic disorders such as Type 2 diabetes and obesity. Furthermore, obesity results in long-lasting changes in the cytoarchitecture and synaptic plasticity of the brain, particularly in the hypothalamus [5,6,7]. Hence, a comprehensive understanding of how the CNS fine-tunes metabolic functions could aid in the further development of therapeutics for various metabolic disorders. A key aspect of current metabolic research is focused on understanding the contributions of the hypothalamic and the brainstem circuitry in the regulation of appetite and energy homeostasis. By manipulating neuronal populations located in these regions, research has uncovered several major neural circuits that exert control over appetite and metabolic functions.

## 2. Hypothalamic and Brainstem Nuclei Involved in Appetite Control and Energy Balance

The hypothalamus and brainstem are critical components of the homeostatic system that regulates appetite and energy balance. These key brain regions have distinct neuronal populations and nuclei, having both complimentary and contrasting roles, exert tremendous control over several facets of energy balance. This occurs both at the level of energy intake and energy expenditure. This section serves to introduce the distinct hypothalamic and brainstem nuclei, and its primary role in the regulation of energy balance. More detailed mechanisms involving the regulation of glycemia and lipid metabolism will be discussed in the next section.

### 2.1. Hypothalamus

The hypothalamus is one of the most well-studied brain regions in metabolism. Apart from regulating a broad range of thermoregulatory, reproductive, and cardiovascular functions, it also exerts tremendous influence on several aspects of energy balance. The hypothalamus is composed of multiple nuclei located adjacent to the third ventricle. These nuclei comprise a distinct subpopulation of neurons, capable of altering energy intake and/or energy expenditure via anabolic or catabolic functions. The complex nexus of hypothalamic neuronal interconnections can integrate responses from peripheral signals (hormones, nutrients, and metabolites), to modulate appetite centrally, and to influence lipid and glucose metabolism, peripherally. Additionally, they also have reciprocal projections to and from extrahypothalamic nuclei located in the brainstem, midbrain, and forebrain which can also alter synaptic activity in the hypothalamic metabolic circuits. Consequently, via integration and coordination of responses from the brain and periphery, hypothalamic nuclei are key regulators of energy homeostasis.

#### 2.1.1. Arcuate Nucleus (ARC)

The ARC is considered as one of the most important brain regions involved in the regulation of appetite and energy expenditure. Located near the median eminence, a region enriched in fenestrated capillaries, the ARC is accessible to circulating hormones, nutrients and metabolites, thus, serving as an ideal relay center to communicate circulating peripheral signals to the brain. The ARC comprises two distinct neuronal subpopulations that have opposing roles in energy homeostasis, the anabolic neuropeptideY/Agouti-related protein (NPY/AGRP) neurons and the catabolic, pro-opiomelanocortin (POMC) and cocaine-and amphetamine-regulated transcript (CART) or POMC/CART neurons (referred to henceforth as AGRP and POMC neurons). Both these neurons are first order neurons, which have glucose and nutrient sensing capabilities, in addition to receiving input from circulating hormones and satiety signals [8,9,10,11]. These counterregulatory neuronal populations are modulated by energy status. Food deprivation rapidly activates AGRP neurons and inhibits POMC neurons [12,13,14]. AGRP neurons release both NPY, which is an agonist for the Y1-5 receptors, and AGRP, an inverse agonist for melanocortin receptors [15,16]. Ablation of AGRP neurons results in a dramatic reduction in feeding, while acute activation results in a robust increase in food intake, weight gain, and altered autonomic outflow to several organs and tissues [17,18,19,20]. NPY was one of the first orexigenic neuropeptides to be identified, and subsequent functional studies revealed a potent, albeit fleeting, appetite-stimulating effect [21]. More recently, it has been revealed that NPY-mediated effects on feeding are mediated via the Y1 receptor, while its effects on energy expenditure are driven via the Y2 receptor [22]. Although both of these orexigenic neuropeptides, NPY and AGRP, have complimentary roles in triggering a hyperphagic response and reducing energy expenditure, the longer-lasting or sustained effect of these neurons on food intake is dependent on AGRP release, while the more rapid effect on food intake is dependent on NPY secretion [20,21,23]. Additionally, AGRP neurons also release GABA, which plays an integral role in AGRP-mediated effects on appetite and energy balance [21,24]. Furthermore, diet-induced obesity blunts AGRP responsiveness to circulating hormones [25]. In stark contrast to AGRP neurons, POMC neurons have a pronounced catabolic effect due to their ability to release the anorectic neuropeptide, α-melanocyte-stimulating hormone (α-MSH), a major satiety neuropeptide which is an agonist of melanocortin receptors [26]. Ablation of POMC neurons was reported to result in a mild obesity phenotype characterized by both reduced and increased food intake [27,28]. Interestingly, only chronic, but not acute chemogenetic activation of these neurons results in suppression of food intake, suggesting a role for POMC in maintaining long-term energy homeostasis [28]. POMC neurons have been reported to exhibit functional and spatial heterogeneity characterized by differences in both molecular architecture and anatomical projections to distinct brain regions, suggesting a more complex neural network involved in metabolic control [29,30,31]. POMC neuronal activity is also regulated by AGRP neurons. Anatomic and functional evidence indicates that GABA-releasing AGRP neurons are involved in inhibiting POMC neuronal activity and α-MSH release [32,33,34]. Apart from α-MSH, it is also to be noted that POMC neurons also release β-endorphin, which binds to the μ-opioid receptors. Both these POMC-derived neuropeptides have functionally antagonistic roles in the regulation of energy balance [35]. Both hypothalamic AGRP and POMC neurons are known to express the μ-opioid receptors (MOR). In the case of POMC neurons, the MORs function as autoinhibitory receptors that are activated by the release of β-endorphins [36]. Interestingly, while α-MSH is predominantly involved in suppressing appetite, β-endorphin were shown to play a major role in promoting a palatability-driven feeding response [37,38]. Naltrexone, a MOR antagonist which has been shown to suppress feeding on a short-term basis, has been shown to have stimulatory effects on POMC neurons in both rodents and humans [39,40]. More about their therapeutic utility will be covered in a later section.

Both AGRP and POMC neurons also express receptors for insulin (IR) and leptin (LepR). Leptin depolarizes and increases firing frequency of POMC neurons, while hyperpolarizing and inhibiting AGRP/NPY neuronal activity and neuropeptide release [41,42,43,44,45]. Mechanistic studies revealed that deletion of Rho-kinase 1, a protein kinase involved in cytoskeletal reorganization and neuropeptide release, in both AGRP and POMC neuronal populations resulted in leptin resistance and obesity [46,47]. Collectively, these data point to a crucial central mechanism by which leptin can induce a negative energy balance. Studies investigating the role of insulin signaling in both AGRP and POMC neurons on appetite regulation have yielded contradictory results. While some studies reported on little-to-no effect on appetite and body weight change with IR deletion in AGRP neurons, others have described a more nuanced role of AGRP-specific insulin signaling on regulating meal size [48,49]. A context-dependent appetite suppression role is reported for insulin signaling in the AGRP neurons, which is characterized by acute repression of feeding bouts without altering total calorie intake, and the suppression of highly palatable high-fat-diet food over standard chow [49]. In the case of POMC neurons, while the deletion of LepR results in mild obesity, knockout of IR in these neurons had no significant effect on body weight [50,51]. Furthermore, both AGRP and POMC neurons are modulated by postprandial signals, such as ghrelin, incretins, and amylin, to regulate food intake [52,53,54,55,56]. Apart from having integral roles in appetite and satiety regulation, these neuronal populations are also involved in maintaining glucose homeostasis as chemogenetic activation of AGRP and POMC neurons revealed distinct roles of G protein activation on food intake and glycemic control [18,20,28,57]. The mechanisms through which both these neuronal populations regulate glucose homeostasis will be discussed in later sections of this review. Additionally, both AGRP and POMC neurons can also regulate energy balance via the hypothalamic–pituitary–thyroid (HPT) axis. HPT axis is well-known to stimulate energy expenditure. Thyroid hormones play an important role in maintenance of homeothermia, and stimulation of the thyroid axis is known to increase energy expenditure via thermogenesis [58]. ICV administration of NPY has been shown to suppress circulating levels of thyroid hormones [59]. Interestingly, the melanocortin system has also been shown to regulate the HPT axis. Both in vivo and in vitro studies revealed that α-MSH can stimulate the HPT axis by increasing the levels of thyroid stimulating hormone (TSH), while AGRP on the other hand inhibits it [60,61]. For more information on the role of the melanocortin system in regulating the HPT axis, readers can refer to other reviews on this topic [62].

Another key aspect of the ARC neurons, especially POMC, is that they exhibit sexual dimorphism. Higher number of POMC neurons and increased neural activity were observed in female animals when compared to their male counterparts [63]. Disruption of key genes in POMC neurons in female mice resulted in the development of obesity [63,64,65,66]. More recently, POMC-specific alteration of certain highly expressed CNS genes, resulted in changes in glucoregulation and energy balance in female mice only [67,68,69].

ARC is highly susceptible to synaptic plasticity in response to the hormonal milieu. Both AGRP and POMC neurons have been described as exhibiting some level of synaptic rewiring under periods of food deprivation and overfeeding conditions [70]. Particularly, the melanocortin system has been reported to exhibit synaptic remodeling under both extreme metabolic changes, such as starvation and overfeeding, but also under physiological feeding states which results in modest metabolic changes [71,72,73]. Plasticity of the ARC has important implications in obesity, as diet-induced obesity has been demonstrated to suppress hypothalamic remodeling and neurogenesis resulting in reduced neuronal turnover [74]. It was also demonstrated to result in reactive gliosis in the ARC with altered synaptic architecture of the NPY and POMC neurons [75]. High fat diet (HFD)-induced neurogenesis is not restricted to the neuronal populations alone in the ARC. HFD activated neurogenesis in the median eminence however leads to energy storage, while prevention of it results in a reduction in weight gain [76]. Stimulation of neurogenesis in response to HFD is observed in female mice and not in males, suggesting a sexual dimorphic nature of hypothalamic neurogenesis [77].

#### 2.1.2. Paraventricular Nucleus (PVH)

The PVH serves as an important convergence/termination point for orexigenic and anorexigenic projections arising from the ARC and other hypothalamic regions. Neurons present in this region express two different types of melanocortin receptors subtypes (MC3R and MC4R) that can be activated by the melanocortin peptide, α-MSH [78,79]. α-MSH and AGRP, released from the ARC projections, can modulate PVH neuronal activity by either activating or antagonizing the melanocortin receptors, respectively [16,79,80,81]. Thus, these neurons provide counterregulatory inputs to fine tune energy balance in response to changes in the levels of circulating signals. PVH neurons express single-minded 1 (Sim1), a transcriptional factor required for PVH development and the maintenance of energy homeostasis [82,83]. Sim1 neurons have pronounced effects on satiety and energy homeostasis as both sim1 heterozygous mice, and inducible Sim1-deficient mice, exhibit hyperphagia leading to obesity [83,84]. A major subset of Sim1 neurons in the PVH express MC4R [50,79]. Mutations in the MC4R gene are a leading cause of monogenic forms of obesity, and MC4R variants have been linked to increased obesity in certain populations [85,86,87,88]. The MC4R/Sim1 neurons, located in the PVH, together with the POMC neuronal projections, arising from the ARC, form the melanocortin pathway in the hypothalamus. Stimulation of MC4R neurons in the PVH results in pronounced satiation effects and thereby can induce a negative energy balance and confer protection against obesity [50,89,90,91]. Interestingly, short-term administration of MC4R agonists can also increase resting energy expenditure and shift substrate utilization towards increased fat oxidation in obese individuals suggesting additional mechanisms through which the melanocortin pathway and Sim1 neurons induce a negative energy balance [92]. Knockdown of MC4R results in potential disruption of synaptic plasticity and attenuation of long-term potentiation in the PVH [93]. Perturbation of MC4R signaling in the PVH alone, or in both PVH and DMV results in hyperphagic obesity with reduced energy expenditure and defects in insulin sensitivity [89,94]. It is to be noted that MC4R-expressing neurons are not just located in the hypothalamic nuclei, but also located in the brainstem, intermediolateral cell column of the spinal cord, and autonomic neurons where they not only exert prominent cardiovascular effects, but also regulate metabolic functions including thermogenesis, glucose homeostasis and energy expenditure [95,96,97,98]. Interestingly, PVH not only comprises MC4R neurons, but also contains other neuronal populations such as prodynorphin-expressing neurons, which lack MC4R. These neuronal populations have comparable effects to the PVH-MC4R expressing neurons on regulating satiety [99]. Several such anatomically distinct neuronal populations have been identified in the PVH as having appetite-regulatory roles, which further highlights the complexity of this nucleus [100,101,102,103]. For a more detailed review on the pathophysiological roles of MC4R neurons, readers can refer to excellent reviews on this topic [104,105].

#### 2.1.3. Ventromedial Nucleus of the Hypothalamus (VMH)

Despite having an inauspicious history in metabolism research, the VMH is still appreciated as one of the principal satiety centers in the brain [106]. Early studies have highlighted an important role of VMH in suppressing appetite [107,108]. Apart from regulating food intake, VMH neurons have also been associated with improvements in several metabolic parameters and conferring protection against obesity [109,110]. A major subset of VMH neurons express steroidogenic factor 1 (SF1), often serving as a biomarker to distinguish VMH from other hypothalamic nuclei. Similar in function to the POMC neurons, activated SF1 neurons elicit pronounced anorexigenic effects with increased energy expenditure [111,112]. These neurons not only provide excitatory input directly onto the POMC neurons, but also project to the paraventricular thalamus to induce an aversive effect and suppress appetite [113,114]. Deletion of LepR from SF1 neurons also resulted in a similar degree of weight gain in mice when compared with LepR-specific KO in POMC neurons, suggesting important roles of leptin signaling in both sets of neuronal populations [50,115]. Moreover, SF1 neurons have distinct projections to other regions of the brain involved in negating insulin-induced hypoglycemia [116]. This will be covered in a later section. For a more detailed review on the role of SF1 neurons in metabolic disorders, the readers can refer to the review by Fosch et al. [117].

#### 2.1.4. Dorsomedial Hypothalamus (DMH)

Another hypothalamic nucleus that affects feeding response is the DMH. DMH lesion in both young and older rats produced a hypophagic response with reduced body weight [118]. Interestingly, the DMH expresses NPY, which shows altered levels in various models of obesity [119,120,121]. Overexpression of NPY in the DMH results in an increase in food intake, weight gain, and an obese phenotype under high-fat-diet conditions, while knockdown of NPY ameliorated these effects in obese mice [122]. Inhibitory GABAergic neurons projecting to the PVH have been proposed as a key mechanism for eliciting a DMH-mediated orexigenic response [123]. Additionally, DMH neurons project to the ARC where they inhibit POMC neurons during fasting suggesting parallel neural circuits from DMH to regulate appetite [14]. The DMH may also be involved in the regulation of food intake by other hormones and peptides, as intra-DMH administration of the appetite-suppressing hormone, cholecystokinin (CCK), resulted in a suppression of food intake [124,125]. Interestingly, under refeeding conditions, excitatory glutamatergic projections are also activated by a subset of DMH glutamatergic neurons leading to reduced food intake [126]. A recently published study reported on DMH having bidirectional effects on food intake, which receive key leptin-responsive projections from the AGRP neurons [127]. Thus, it is likely that DMH projections could participate in the fine tuning of energy intake by activating distinct inhibitory and excitatory projections to other hypothalamic nuclei.

### 2.2. Brainstem

The brainstem exerts significant control over autonomous biological functions. The medulla is a key brainstem structure which has prominent cardioregulatory and metabolic functions, via specialized cardiovascular and satiety centers, respectively. The medullary cardiovascular centers have well-established roles in the homeostatic regulation of blood pressure via the baroreflex [128]. Although the brainstem is not as well-investigated as its counterpart, the hypothalamus, in metabolism, studies dating back to the 1970s highlighted the importance of the caudal brainstem in mediating satiety and glucoregulatory responses [129,130]. Importantly, specialized medullary regions serve as crucial integration points between the CNS and the digestive tract. They receive visceral afferent input from gastrointestinal sensory neurons, the latter conveying satiety signals in response to a meal. Additionally, the brainstem also comprises a circumventricular organ, area postrema (AP), which allows access to satiety signals. These signals in turn can modulate adjacent and supra-adjacent neuronal populations located in the brainstem. These proximally located neuronal populations in the caudal brainstem, in conjunction with the AP, are key structures in mediating postprandial satiety [131,132].

#### Dorsal Vagal Complex (DVC)

The caudal brainstem not only expresses receptors for circulating pressor peptides, but it can also be modulated by metabolic cues and thus exerts control over energy homeostasis [133,134,135,136,137]. The DVC located in the hindbrain is designated as the brainstem satiety center. The DVC comprises the AP, the nucleus of the solitary tract (NTS), and the dorsal motor nucleus of the vagus nerve (DMV). The NTS serves as the primary hub for ascending neural signals from the nodose ganglia, which contains cell bodies for several vagal afferents that densely innervate the gastrointestinal tract (GIT) [138]. The sensory vagal nerve terminals in the GIT are heterogenous in nature conveying both chemosensory, from nutrients and gut hormones, and mechanosensory signals to the brainstem [138,139]. Postprandial gut hormones and nutrients suppress food intake by transmitting information via the sensory vagal afferent terminals to the NTS, a crucial entry point in the brain for visceral information [140,141]. CCK, one of the first gut peptides to be identified to mediate satiety, elicits its actions by acting on the CCK-A receptors that are abundantly expressed on the vagal afferents and the cell bodies of the nodose ganglia [140,142,143,144]. Additionally other receptors involved in regulating satiety, such as the LepR, are also expressed on these cell bodies [145]. As a result, circulating signals such as leptin can also act along with CCK on the nodose ganglia, to synergistically suppress food intake [146,147]. Another class of gut hormones, the incretins, also exert prominent effects on satiety and glucose homeostasis. The incretins, glucagon-like peptide-1 (GLP-1) and glucose-dependent insulinotropic polypeptide (GIP), are released in response to a meal, and they act on their corresponding receptors (GLP1R and GIPR) located in pancreatic islets where they promote insulin release. These receptors are also expressed in non-islet cells, where they exert prominent metabolic actions independent of direct effects on pancreatic insulin secretion [148]. In the GIT, the GLP1R is expressed on mechanosensitive vagal sensory neurons, and its activation results in pronounced inhibition of food intake, while its knockdown is associated with an increased meal size [149,150].

Interestingly, profiling of G-protein coupled receptors in the GIT revealed a lack of receptor expression for GIP and ghrelin on vagal afferents, potentially highlighting other CNS-dependent mechanisms to alter food intake [151]. Similar to the ARC, the AP lacks a well-defined blood–brain barrier, as a result is accessible to various satiety signals and circulating hormones. Since the NTS is located in close proximity to the fourth ventricle, it serves as a crucial node for integrating signals from the gut and circulation. Satiety signals, such as CCK, GLP-1, and their analogs, have been shown to inhibit appetite by acting on their corresponding receptors localized to the brainstem neurons in the satiety center [152,153,154]. The GLP1R is highly expressed in the NTS, and knockdown of preproglucagon, a precursor for GLP1, in the brainstem results in hyperphagia and increased adiposity, suggesting a crucial role for central GLP1R in mediating satiety [155]. Interestingly, GIPR agonism not only enhances the anorectic effect of GLP1R agonism, but recent studies suggest that it improves tolerability of PYY analogs by modulating the brainstem neural circuits and blocking its anorectic effect [156,157,158]. Thereby, understanding of incretins-mediated modulation of the brainstem neural circuitry has significant implications for the development of weight loss drugs with an improved side effect profile. The role of incretins in the hypothalamic and brainstem neural circuitry in the regulation of energy homeostasis will be discussed in a later section. Other pancreatic and gut-derived postprandial signals, such as amylin and PYY, also act on neuronal populations in the AP and NTS to promote satiety [159,160,161,162,163]. Additionally, leptin-mediated signaling in the NTS also activates the satiation neural circuitry to suppress food intake and regulate energy balance [164,165,166].

Projections from the NTS extend to other brain regions involved in appetite control and food aversion behaviors, where they suppress appetite by triggering either a positive or negative valence [167,168]. The latter may well be dependent on both the molecular architecture of the neural circuit, and the brain regions innervated by it. For instance, NTS projections to calcitonin gene-related protein (CGRP) expressing neurons located in higher brain regions, are strongly involved in mediating anorexia and reducing body weight [169,170]. However, they can exert opposing motivational valences, since projections to specific brain regions can generate both a positive valence (NTS to PVH projection) and a negative valence (NTS to PBN projection); the latter aversive response triggered by the activation of CGRP neurons in the PBN [171,172,173]. In stark contrast to the CCK neurons, calcitonin receptor expressing neurons from the NTS do not activate CGRP neurons, and hence produce a non-aversive suppression of food intake despite projecting to the PBN [173]. Other neuronal populations such as GLP-1 expressing neurons, which are primarily located in the caudal NTS, have projections to the VTA where they regulate intake of highly palatable food [174,175].

The NTS neurons also comprise a small, but metabolically relevant, population of POMC expressing neurons, accounting for about 10% of the total POMC neuronal population [176,177]. Interestingly, while they are activated by postprandial visceral afferents from the gut, they do not co-express several of the other neuropeptide markers observed in the NTS, suggesting a distinct hub of neurons involved in mediating satiety [178,179]. These neuronal populations are functionally similar to the POMC neurons in the ARC, but they exhibit different kinetics in terms of suppression of food intake. ARC-POMC neurons are involved in long-term suppression, while the NTS-POMC neurons mediate short-term feeding responses [28]. The latter neurons are potentially involved in a more rapid feeding suppression via circulating satiety signals. NTS-POMC neurons have been shown to be crucial for the acute appetite-suppressing effect of lorcaserin, indicative of their clinical relevance [180]. More recently, this effect of lorcaserin was also shown to be meditated via the GLP-1 neurons in the brainstem, in addition to the NTS-POMC neurons [181].

In addition to the regulation of food intake, the hindbrain circuitry also has important roles in glucose sensing and modulation of systemic glucose via vagal efferents [182,183]. Neuropeptide FF (NPFF), a key analgesic peptide which has been demonstrated to have a role in substrate utilization and regulation of energy balance, is strongly expressed in the caudal brainstem, mainly localized in the DVC [184]. More recently, a study reported on impairments in glucose homeostasis in mice deficient in NPFF, further highlighting the glucoregulatory role of the DVC [185]. The role of the various satiety signals in regulating glucose and lipid metabolism via brainstem circuits will be covered in more detail in later sections.

The sections so far highlight the pivotal roles of hypothalamic orexigenic and anorexigenic neuronal populations, along with the brainstem satiety center, in the regulation of energy intake and expenditure. A schematic summarizing this is shown in Figure 1. While the NTS integrates multiple metabolic cues to promote satiation, the ARC neuronal populations are able to exert both short- and long-term effects on energy homeostasis in response to energy demands.

## 3. Crosstalk between Hypothalamic and Brainstem Nuclei with Metabolic Organs to Regulate Energy and Glucose Homeostasis

Autonomic dysfunction is associated with an elevated risk of developing metabolic syndrome and cardiovascular diseases [128,186]. In the case of metabolic disorders, this is due to an augmentation of sympathetic activity resulting in a breakdown of the glucose homeostatic processes [187]. The end result is a chronic elevation in blood glucose due to an imbalance between glucose production and glucose clearance from the blood by insulin-sensitive organs. The resulting hyperglycemic condition is known to result in extensive vascular complications due to endothelial damage, and hence serves as an independent risk factor for cardiovascular diseases [188,189]. Understanding the potential mechanisms involved in the maintenance of glucose homeostasis is therefore of high clinical relevance. Hypothalamic and brainstem nuclei are key components of a central network that help maintain a balance between sympathetic and parasympathetic nerve activity to the endocrine organs, resulting in exerting significant control over glucose metabolism. Circulating signals act on these neurons to recalibrate autonomic efferents to the peripheral metabolic organs. This section focusses on the crosstalk between the various brain regions discussed in the preceding section, and the important metabolic organs involved in maintenance of glucose and energy homeostasis.

### 3.1. Brain-Pancreas Axis and the Role of Central Insulin Signaling

Multiple brain regions, including several hypothalamic and brainstem nuclei, contain neural networks that regulate pancreatic islet function via autonomic efferents [190,191,192]. Functional validation of these circuits revealed important roles for several hypothalamic nuclei in pancreatic insulin release [192]. These hypothalamic nuclei exhibit bidirectional control over insulin release. Stimulation of a subpopulation of oxytocin neurons in the PVH suppressed insulin secretion, whereas increased glucokinase activity in the ARC augmented glucose-stimulated insulin secretion and improved glucose tolerance [193,194]. In addition to the hypothalamic nuclei, pancreas-projecting DMV neurons were reported to be excited by GLP1, which can then potentially increase insulin release by a vagal efferent pathway [195,196].

IR is widely expressed in the brain, which enables circulating insulin to modulate neuronal populations that are integral to metabolism [197]. The hypothalamus represents a crucial insulin-responsive brain region involved in maintaining euglycemia [198]. The hypothalamic IR signaling and a downstream target of IR, the K-ATP channels, have both been reported to have essential roles in the regulation of endogenous glucose production [199,200]. ICV infusion of insulin into the third ventricle suppresses endogenous glucose production, while perturbation of central insulin signaling impaired both glucose and lipid metabolism [48,201]. Interestingly, the divergent mechanisms through which central insulin signaling regulates glucose and lipid homeostasis may be attributed to differing outcomes of IR activation in AGRP versus the POMC neurons. Insulin effects on AGRP neurons result in improved glucose homeostasis, while its action on POMC results in changes in lipid metabolism [202]. However, these effects may be more nuanced, as insulin receptor signaling in the POMC neurons has been shown to regulate glucose homeostasis which is shown to depend on the nutritional (fed vs. fasted) and the pathophysiological status (obese vs. lean) of the mice [203]. A similar subtle, yet significant, effect was observed for insulin-mediated feeding suppression via the AGRP neurons, which is described in an earlier section. More recently, the antihyperglycemic effect of hypothalamic insulin signaling was shown to be dependent on the neuropeptidegric system, 26Rfa and its receptor GPR103 [204]. ICV administration of both insulin and 26Rfa greatly augmented glucose-mediated insulin release, and GPR103 blockade greatly suppressed both their effects on glucose homeostasis, suggesting a potential key mechanism of central insulin signaling [204].

In addition to the ARC, insulin signaling in the brainstem has also been investigated. IR activation in the brainstem nuclei modulates both food intake and glucose production via distinct intracellular signaling complexes [205,206]. Furthermore, insulin has been shown to decrease synaptic activity in the DVC by hyperpolarization, to potentially alter gastric function [207,208]. While insulin mostly suppresses excitatory neuronal activity in the DMV, under certain conditions of elevated cAMP levels, it was able to suppress inhibitory neurotransmission in only normoglycemic, but not hyperglycemic mice, suggesting a mechanism of potential pathophysiological relevance [209]. Apart from circulating insulin being able to modulate hypothalamic and brainstem neuronal activity, there is evidence of insulin being produced locally in the brain as well [210,211]. While insulin producing hypothalamic neurons were shown to have both anabolic and catabolic roles, in the case of brainstem a recent study highlighted an anabolic role for them [212,213,214]. All of the aforementioned studies highlight the importance and diversity of hypothalamic and brainstem insulin signaling in metabolic control.

### 3.2. Brain–Liver Axis

The liver is a major site of glucose metabolism, by promoting both glucose production via gluconeogenesis and glycogenolysis, and stimulating storage via glycogenesis. These hepatic metabolic pathways are under the strict control of circulating hormones and hepatic autonomic efferents. While sympathetic innervation enhances glucose production, the vagal branch has been shown to inhibit glucose production and promote storage [215,216].

As described earlier, central insulin signaling plays a key role in improving glucose homeostasis. To a large extent, insulin modulates the hypothalamic neural circuitry to regulate autonomic efferents to the liver. Insulin activates the hypothalamic K-ATP potassium channels resulting in diminished hepatic gluconeogenesis via modulation of vagal efferent activity [217]. Activation of K-ATP channels is known to result in neuronal hyperpolarization and subsequent reduction in the release of neuropeptides [218]. In line with these findings, abrogation of neuropeptide release from AGRP neurons (both NPY and AGRP) has been described as a crucial mechanism by which insulin markedly alters hepatic efferents, both sympathetic and parasympathetic, to suppress hepatic glucose production [48,51,219,220].

In addition to AGRP, the role of POMC neurons in hepatic glucose control via both insulin and leptin signaling has been explored in the brain. Conflicting reports have emerged regarding the role of POMC-specific insulin signaling in altering hepatic gluconeogenesis. While some studies have reported that hepatic gluconeogenesis is mostly under the control of AGRP-specific and not POMC-specific insulin signaling [51,202], other investigators have concluded that POMC-insulin signaling plays a key role in suppressing hepatic glucose production [203]. The differences noted by various research groups may be attributed to the POMC neuronal heterogeneity which is described in an earlier section. Interestingly, both insulin and leptin can depolarize and also hyperpolarize a subset of POMC neuronal population, which could also contribute to the differences observed in the glucoregulatory outcomes observed with central insulin signaling [221,222,223]. POMC-specific leptin signaling has been demonstrated to improve glucose homeostasis, independent of its effects on food intake and appetite, via improvements in hepatic insulin sensitivity [224,225]. Interestingly, ICV leptin infusion does not alter glucose production in the liver, but triggers striking alterations in hepatic glucose fluxes [226,227]. Other hypothalamic nuclei, such as the SF1 neurons in VMH, also contribute to the regulation of hepatic glucose production. Stimulation of SF1 neuronal projections to specialized basal forebrain structures counteracts hypoglycemia by increasing blood glucose [116,228]. The VMH neurons maintain euglycemia in energy deprived states by regulating the expression and activity of hepatic gluconeogenic and glycogenolytic genes [112,229]. Therefore, VMH neurons may activate distinct neural circuits under glucopenic conditions to elevate endogenous glucose production. It is to be noted that VMH neurons exhibit neuronal heterogeneity and activation of SF1 neurons has been linked to hyperglycemic responses characterized by insulin resistance [114,230]. Apart from the VMH, hyperactivity of liver-projecting PVH neurons has also been reported in a diabetic mouse model [231].

The hypothalamic neural circuitry does not work in isolation to regulate endogenous glucose production. Glucoregulatory neural circuits between the hypothalamus and brainstem have been reported to regulate hepatic glucose production under both hyperglycemic and hypoglycemic conditions [232,233]. As mentioned earlier, neurons located in the DVC serve as an important integration point for ascending signals from the gut, as well as signals from other regions of the brain. Activation of NMDA receptors in the DVC lowered glucose production via hepatic vagal efferents [234]. Furthermore, administration of an NMDA blocker into the NTS blocked the hepatic glucose lowering effects of intestinal lipids [235]. This suggests an integral role of the brainstem neurons in nutrient-mediated changes in glucose homeostasis by regulating hepatic glucose production. A more detailed understanding of the hypothalamic and brainstem neural circuits in the regulation of hepatic efferents under both physiological and pathological conditions could aid in better understanding the mechanisms underlying impaired glucose homeostasis in metabolic disorders.

### 3.3. Brain–Adipose Tissue Axis

A key metabolic tissue that has prominent roles in glucose and whole-body energy homeostasis is the adipose tissue. Chemogenetic and optogenetic modulation of adipose tissue has dramatic metabolic effects in both lean and obese conditions [236,237,238,239]. Hypothalamic and brainstem neuronal populations regulate key autonomic projections (mainly sympathetic) from the CNS, to both white and brown adipose tissues (WAT and BAT, respectively) [240,241,242,243]. The best-defined neural circuits for adipose tissue regulation involves the AGRP and POMC neurons of the ARC. AGRP neuronal stimulation not only modulates hepatic glucose production, but also contributes to insulin resistance by inhibiting glucose uptake from BAT [244]. AGRP stimulation also alters substrate utilization in adipose tissue shifting its energy source towards carbohydrates and away from lipids. This effect involves decreasing fat oxidation and increasing lipogenesis resulting in increased adiposity [245]. Interestingly, the role of orexigenic neuronal populations in the regulation of adipose tissue function is not limited to the ARC. NPY knockdown in the DMH resulted in a favorable metabolic profile characterized by an increased BAT mass and enhanced beiging of WAT, leading to increased thermogenesis and energy expenditure [246]. In contrast to the AGRP neurons, perturbation of POMC neuronal activity by either knocking out IR or by genetic inactivation of key mitochondrial proteins, results in an altered adipose tissue lipolytic profile, which potentially contributes to high-fat-diet-induced metabolic impairments [202,247]. In addition to the ARC, other hypothalamic nuclei could also be involved in regulating lipid metabolism. There is evidence that hypothalamic AMPK, a metabolic regulator activated by low energy states, plays a role in glucose homeostasis via modulation of sympathetic outflow to adipose depots [248]. Specifically, AMPK in VMH has been linked to thermogenesis and beiging of WAT [249,250].

Leptin is a key adipokine released by WAT, and acts on the LepR expressed in the CNS to induce a negative energy balance. The LepR which is widely expressed in the CNS, plays a key role in modulating the neural circuits involved in regulating autonomic outflow to adipose tissue, thus modulating lipid metabolism in the adipose depots [251,252]. Potentiation of leptin and insulin signaling in POMC neurons confers protection against diet-induced obesity by increased WAT browning and decreased adiposity [253]. While activating LepR in key brainstem nuclei regulates sympathetic outflow to kidney [254,255], activation of the hypothalamic LepR alters metabolism via regulation of sympathetic outflow [251,256]. Furthermore, pancreatic peptides have also been shown to act in the brain to alter adipose tissue function. For instance, perturbation of amylin/calcitonin signaling in POMC neurons results in increased adiposity and decreased UCP1 in BAT, resulting in impaired glucose tolerance [257]. Therefore, via the hypothalamic and brainstem neural circuits, various aspects of adipose tissue functionality could be fine-tuned to have a sizable impact on whole-body energy homeostasis.

The sections so far highlight the contrasting roles of AGRP and POMC neuronal populations in glucose and lipid homeostasis. Figure 2 describes the key mechanisms involved in mediating these effects.

### 3.4. Gut–Brain Axis and the Role of Incretins

Neuronal populations located in the hypothalamus and brainstem express receptors for gut hormones. As discussed in earlier sections, the enteroendocrine system of the gut is responsible for chemosensing, and thus regulates the release of gut hormones in response to a meal to trigger a satiation response [258]. The incretins, GIP and GLP1, are one such class of gut hormones that are not only capable of suppressing appetite, but also have significant effects on maintenance of postprandial glucose levels. They exert prominent metabolic and glucoregulatory roles via their receptors expressed both in the periphery and the CNS [259]. In response to a meal, both GIP and GLP1 levels are elevated in the blood, which affects appetite and energy homeostasis by modulating neural activity in key hypothalamic and brainstem nuclei. The incretin receptors, GLP1R and GIPR, are highly expressed in the ARC and DMV, and thereby can be activated by circulating incretins and their analogs to regulate food intake and energy balance [260,261,262,263,264,265]. Multiple studies have highlighted a crucial role of the ARC in mediating the appetite suppressing effects of incretins [266,267,268,269]. The appetite-reducing mechanism of the GLP1 analog, liraglutide, involves activation of GLP1R in ARC, specifically activation of the anorectic POMC neurons [262]. Additionally, CCK neurons in the NTS also play crucial roles in mediating the full anorectic effect of GLP1R agonists, but not GIPR agonists [270]. In the case of GIP, hypothalamic GIPR-expressing neurons are reported to have essential roles in mediating the effects of GIP on feeding and energy homeostasis [263,271]. Chemogenetic activation of GIPR in the hypothalamus, and modulation of hypothalamic neuronal activity by peripheral GIPR agonists resulted in an inhibition of food intake and improved glucose handling [263,264]. Studies have also reported on the beneficial role of activating, and not antagonizing, GIPR in promoting weight loss in diet-induced obesity and improved glucose homeostasis [272,273]. More recently, acute administration of a long acting GIPR agonist, GIPFA-085, acted via the ARC POMC neurons to suppress feeding and increase lipid utilization, while subchronic administration was shown to reduce body weight in diet-induced obesity mice [274]. However, there is ambiguity on whether activation or inhibition of GIPR has beneficial effects on obesity. There is evidence highlighting a positive correlation between elevated GIP levels and high-fat diet feeding [275,276]. Furthermore, both global and CNS-specific GIPR deletion resulted in protection against obesity, suggesting that they have essential roles in induction of weight gain and adiposity [264,277]. While it is apparent that central GIPR has a role in energy homeostasis, the relative contributions of the various neuronal populations in mediating the metabolic effects of GIPR are yet to be fully mapped out.

It is well-established that incretins improve glucose homeostasis by augmenting insulin secretion, following activation of their receptors on the beta cells of the pancreatic islets [278,279]. However, incretins are rapidly degraded once released into the GIT and the blood stream, suggesting the existence of a vagal afferent neural pathway as an intermediary mechanism to mediate low dose effects of incretins on regulating glucose homeostasis [149,280,281]. In agreement with this concept, nutrients and other gut hormones have also been shown to trigger the gut–brain axis to regulate glucose and energy homeostasis [235,282,283]. The incretin receptors in the hypothalamus and brainstem have been shown to have important roles in improving glucose homeostasis. Antagonizing the GLP1R located in the ARC resulted in worsening of glucose tolerance, while direct administration of GLP1 into the ARC reduced glucose production [284]. Furthermore, central GLP1R-mediated improvement in glucose homeostasis is preserved under high-fat feeding conditions [285]. GLP1R-expressing neurons are also present in other hypothalamic nuclei. Stimulation of DMH-GLP1R resulted in a reduction in blood glucose via descending input to the DMV, which in turn augments pancreatic insulin release [286]. As discussed earlier, modulation of GLP1R activity in the DMV alone also regulates pancreatic autonomic efferents, and exerts influence over insulin secretion [195,196]. These studies highlight the fact that both peripheral and central incretin receptors act in concert to trigger metabolic improvements observed with the incretins and their analogs. However, while glucose lowering ability of central GLP1R has been mostly reported, there have been reports of hyperglycemic responses by high doses of GLP1R agonist, exendin-4 [287]. In line with this, ICV administration of GLP1 also paradoxically reduced glucose-stimulated insulin secretion and caused mild glucose intolerance [288]. However, both these effects are a consequence of the sympathetic nerve activity activation, and could be due to the activation of GLP1R in several distinct neuronal populations present in the various hypothalamic nuclei. This could also suggest a negative feedback mechanism by which central GLP1R localized on distinct neuronal populations limits insulin release. Further investigation is needed to explore this aspect. Apart from incretins, other postprandial signals have also been involved in regulating glucose and lipid metabolism via the CNS. For instance, FGF19, a postprandial enterokine that has hypoglycemic effects, elicits its effects by modulating both hypothalamic and brainstem neuronal populations to improve glucose homeostasis [289,290,291,292]. Amylin’s effects on food intake and body weight were demonstrated to be dependent on modulation of brainstem neuronal signaling, specifically lateral dorsal tegmental nucleus, resulting in increased SNS activity to BAT [293].

The CNS, specifically the hypothalamic and brainstem neural circuits, regulates several different facets of energy balance by altering autonomic outflow to multiple metabolic tissues and endocrine glands. A schematic summarizing some of the important complimentary and distinct metabolic roles of the neural pathways discussed in this review, is shown in Figure 3.

## 4. Clinical Implications

Multiple clinical studies have underscored the importance of the appetite/satiety centers in the brain for mediating the effects of gut hormones and circulating peptides on energy homeostasis [294,295]. Weight-loss drugs act by inducing a negative energy balance by either central or peripheral mechanisms, or a combination of both. This mainly includes appetite reduction via the melanocortin system, increased energy expenditure via both central and peripheral mechanisms, or restriction of calorie absorption from the intestine by acting on the intestinal enzymes [296]. Several of the anti-obesity drugs in the past were effective in inducing significant weight loss, however they were discontinued due to adverse effects. For instance, rimonabant improved several metabolic parameters along with promoting anorexia in several clinical studies, but it was associated with high neuropsychiatric adverse effects leading it to be withdrawn from the market [297]. Other drugs such as aminorex and sibutramine have been discontinued due to severe cardiovascular events [296,298]. The currently approved anti-obesity drugs demonstrate not only equivalent therapeutic efficacy, but also have favorable cardiovascular and neurological profile [296]. Centrally acting drug combinations such as naltrexone + bupropion, which act on the melanocortin system via the opioid receptors, have been shown to be efficacious in reducing body weight without any CNS adverse effects [299]. Drugs such as orlistat act exclusively by inhibiting fatty acid absorption from the gut [300], while GLP1 analogs such as semaglutide act via multiple mechanisms which encompass both central and peripheral mechanisms to suppress appetite, improve glucose and lipid metabolism, and delay gastric emptying [301,302]. While these drugs have a much-improved cardiometabolic risk profile compared with the previous generations of weight loss medications, the long-term risk profile remains an outstanding question [303]. Additionally, gastrointestinal side effects such as nausea and diarrhea are commonly observed, and may diminish patient compliance which further limits their long-term efficacy [296]. Future research should be geared towards evaluating long term risk-benefit profile, using combination therapy with reduced doses to avoid GIT side effects. Deciphering the neural circuits that suppress appetite without triggering aversive responses could also aid in developing drugs with a favorable risk-benefit profile. Interestingly, both orexigenic and anorectic neuronal populations exhibit sexual dimorphism in metabolism and glucoregulation [63,304,305]. A better understanding of gender differences in energy and glucose homeostasis should aid in developing tailored therapeutic strategies for the treatment of obesity [306].

## 5. Conclusions

Obesity has been long considered to be at epidemic proportions globally, and is both a significant health and economic burden [307,308]. It is now evident that the brain is at the apex of the whole-body energy homeostatic machinery. Our understanding of the hypothalamic and brainstem neuronal circuits has already aided in the development of highly efficacious anti-obesity drugs. While several of these neuronal populations exhibit overlapping metabolic roles, recent studies have brought to light distinct and contrasting mechanisms, thereby enabling the CNS to fine tune metabolic functions under physiological conditions. It is important to note that metabolic disorders are highly heterogenous with distinct metabolic profiles [309,310]. A deeper understanding of the molecular architecture of neuronal populations could aid in exploring multiple drug targets, potentially even tailor-made for the treatment of a specific metabolic profile. Such personalized therapies are already employed for several other pathological conditions; hence, it is feasible that this goal can be achieved for the future treatment of obesity and related metabolic disorders [311].

## Figures and Tables

**Figure 1 cells-12-01801-f001:**
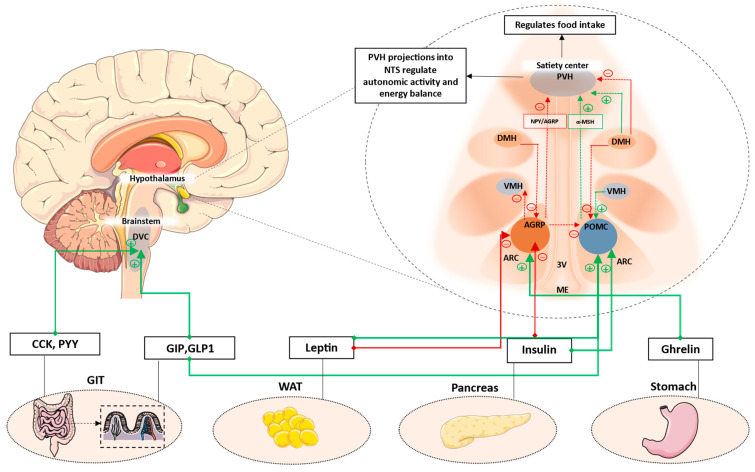
Key hypothalamic nuclei involved in the regulation of appetite and energy balance. ARC, comprising AGRP and POMC neurons, is located next to the median eminence. This region comprises permeable capillaries, thereby allowing access to circulating signals. These signals can modulate ARC neuronal populations, which then have extensive projections to PVH and other hypothalamic nuclei. PVH is the major hypothalamic satiety center. POMC neurons activate MC4R neurons in the PVH to decrease appetite, while AGRP neurons inhibit PVH-MC4R neurons to increase appetite. Additionally, AGRP neurons also inhibit POMC neurons via stimulation of inhibitory GABAergic input to POMC neurons. Anorexigenic signals such as leptin and GLP1 increase satiety by acting on POMC neurons, whereas orexigenic signals such as ghrelin can increase appetite by acting on AGRP neurons. Other hypothalamic neuronal populations have extensive projections to and from adjacent nuclei. While DMH has predominantly inhibitory projections to PVH and POMC, it also has been shown to also have activate inhibitory GABAergic neurons projecting to the AGRP neurons in the ARC. VMH mainly has excitatory projections to the POMC neurons, while AGRP neurons has inhibitory projections to VMH. Additionally, postprandial satiety signals from the enteroendocrine cells of the GIT can also act on DVC located in the brainstem to suppress appetite. AGRP: Agouti-related protein; POMC: Pro-opiomelanocortin; ARC: Arcuate nucleus; ME: median eminence; VMH: Ventromedial nucleus of the hypothalamus; DMH: Dorsomedial hypothalamus; PVH: Paraventricular nucleus; 3V: Third ventricle; DVC: Dorsal vagal complex; CCK: cholecystokinin; GIP: Glucose-dependent insulinotropic polypeptide; GLP1: Glucagon-like peptide-1; WAT: White adipose tissue; GIT: Gastrointestinal tract. Green dotted lines/arrows represent activation. Red dotted lines/arrows represent inhibition. Please refer to Section 2 for more details on the hypothalamic and brainstem neural circuits.

**Figure 2 cells-12-01801-f002:**
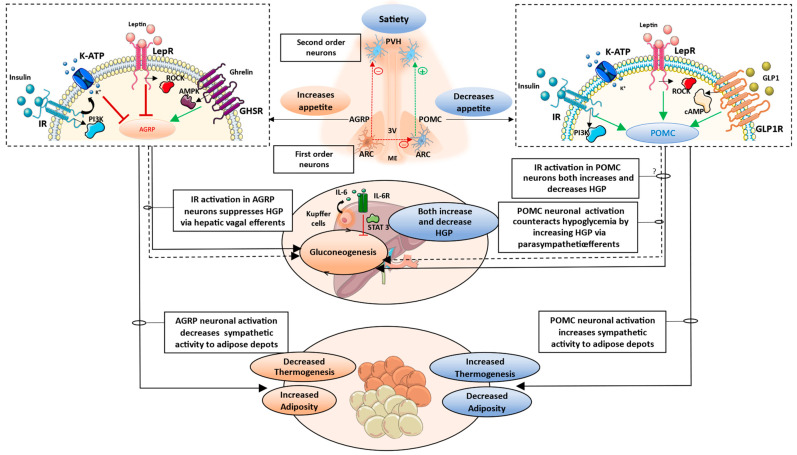
Contrasting roles of AGRP and POMC neurons in glucose and lipid homeostasis. AGRP neurons are depolarized by orexigenic hormones such as ghrelin and hyperpolarized by insulin and leptin. AGRP neurons have integral roles in increasing hepatic glucose production, while its inactivation results in a suppression of endogenous glucose production from the liver. Activation of IR and LepR hyperpolarize AGRP neurons, while GHSR activation depolarizes AGRP neurons. Insulin receptor activation in AGRP neurons results in hyperpolarization via K-ATP channels and a decrease in AGRP neuronal activity. This inturn alters hepatic vagal efferents to reduce endogenous glucose production. This mechanism involves regulation of IL6 release from the liver. Briefly, a decrease in AGRP neuronal activity results in an increase in IL6 release from Kupffer cells. Apart from having inflammatory roles, IL6 also mediates hypoglycemic effects. IL6 acts in a paracrine manner on hepatocytes to suppress hepatic gluconeogenic gene expression, resulting in a reduction in endogenous glucose production. Apart from regulating parasympathetic efferent outflow to the liver, AGRP activation also results in reduced sympathetic outflow to adipose depots. This results in metabolic impairments characterized by a shift in substrate utilization for energy production. Mainly, a lesser reliance on lipid as a predominant energy source, and reduced thermogenesis in BAT. ROCK1 is an important mediator of leptin’s effects in AGRP neurons. Loss of ROCK1 in AGRP has been shown to attenuate leptin’s effect and result in leptin resistance. In contrast to AGRP neurons, IR and LepR have been predominantly shown to depolarize POMC neurons. However, due to POMC neuronal heterogeneity, activation of POMC-specific insulin signaling pathway has been shown to both reduce and increase hepatic glucose production. Incretins such as GLP1 have been shown to act on ARC neurons, predominantly via the GLP1R in POMC neurons. GLP1R activation stimulates POMC neurons resulting in an improved metabolic phenotype. POMC activation results in augmented sympathetic activity to adipose depots, leading to increased thermogenesis and reduced adiposity. Thereby POMC neuronal activation can improve metabolic parameters via the brain–adipose tissue axis. AGRP: Agouti-related protein; POMC: Pro-opiomelanocortin; ARC: Arcuate nucleus; ME: median eminence; PVH: Paraventricular nucleus; IR: Insulin receptor; LepR: Leptin receptor; GHSR: Growth Hormone Secretagogue Receptor; PI3K: phosphoinositide 3-kinase; ROCK: Rho-associated kinase; AMPK: AMP-activated protein kinase; GLP1R: Glucagon-like peptide-1 receptor; cAMP: Cyclic adenosine monophosphate; HGP: Hepatic glucose production; IL6: Interleukin 6; STAT3: Signal transducer and activator of transcription 3. Green dotted lines/arrows represent activation. Red dotted lines/arrows represent inhibition. Black dotted lines represent parasympathetic innervation. Black solid lines represent sympathetic stimulation.

**Figure 3 cells-12-01801-f003:**
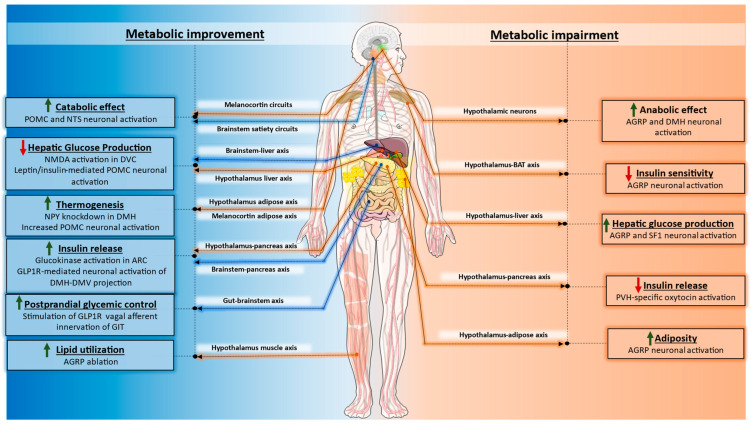
Overview of hypothalamic and brainstem control of energy, glucose, and lipid homeostasis. Both hypothalamic and brainstem neuronal populations exert prominent homeostatic control over energy intake and energy expenditure. Circulating signals can also activate/inhibit distinct neural pathways to recalibrate autonomic outflow to various metabolic organs and endocrine glands. This crosstalk between the CNS and the periphery is essential for the maintenance of glucose and lipid homeostasis. It should be noted that the figure highlights only a limited number of metabolically relevant pathways due to space restrictions. Please see text for details. AGRP: Agouti-related protein; POMC: Pro-opiomelanocortin; NTS: nucleus tractus solitarius; DMH: Dorsomedial hypothalamus; PVH: Paraventricular nucleus; DVC: Dorsal vagal complex; DMV: Dorsal motor nucleus of the vagus; ARC: Arcuate nucleus; NPY: Neuropeptide Y; SF1: Steroidogenic factor 1; NMDA: N-methyl-D-aspartate; GLP1R: Glucagon-like peptide-1 receptor; GIT: Gastrointestinal tract. Orange dotted lines/arrows represent modulation of hypothalamic neural circuits. Blue dotted lines/arrows represent modulation of brainstem neural circuits.

## Data Availability

Not applicable.

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
