# Peer review of "Neurochemical Basis of Inter-Organ Crosstalk in Health and Obesity: Focus on the Hypothalamus and the Brainstem"

_cells, 2023, doi:10.3390/cells12131801_

Round 1
Reviewer 1 Report
This manuscript focuses on the key importance of the brain towards energy metabolism in health and obesity. It is easy to read, and the illustrations are gorgeous. The bibliography is complete so far as the hypothalamus and NTS/DVC are concerned. The chapters on the brain-XXX axis are fascinating since they are not so common in various former reviews. However, the reviewer is uncomfortable with the overall presentation of the review since it completely obliviates 99% of the brain. This is an unfortunate way of thinking common in the rat/mice research community primarily (yet not exclusively) because of the difficulties in accessing the whole rodent brain at an adequate resolution. On the contrary, using larger animal models or even the actual human brain, there is now a vast array of research demonstrating the significant role of the brain as a whole. Furthermore, there is now ample data about IR in the brain itself, somewhat opposite to peripheral IR. Finally, brain network plasticity as a consequence and vice versa as drivers for metabolic imbalance are also major targets for ongoing research. All these are missing in this manuscript.
Author Response
We are grateful to the extremely valuable input provided by the reviewer. These are excellent suggestions which we have incorporated into the manuscript. Also, we appreciate the kind words. Since the initial title was too broad, the reviewer’s comment on the issue of focusing only on certain parts of brain, is apt. In order to better represent the information that is discussed, we have now changed the title to highlight the central aspect of the review paper, that is the inter-organ crosstalk in obesity. Hence, the hypothalamus and brainstem become pivotal regions for our review paper, since these two regions have fenestrated capillaries and thereby serve as entry points for circulating signals from peripheral organs.
We have however broadened the review to include more information as per the reviewer’s suggestions. The point on neuronal plasticity is something we very much liked. We have added more information on this. We have mentioned the importance of it in the introduction itself (lines 37-39; references 5-7). Specifically, we focused on neuronal plasticity pertaining to the ARC (lines 192-206, references 70-77 were added). Neuronal plasticity relating to the PVH was already included in the earlier version (line 234). Regarding IR in the brain, we have covered central insulin signaling in detail in sections 3.1 and 3.2 in relation to its metabolic roles. We have also added a section on HPT axis (lines 175-185).
Reviewer 2 Report
The authors crafted a well structured and articulated review of the central and peripheral neuronal pathways that regulate appetite and energetic homeostasis. The authors also included potential therapeutic applications in the recent developments of reported molecular mechanisms and proposed new/alternative suppositions on the functional outcomes of reported mechanisms. The figures were helpful with the visualization of the neuropathways and affected organs. Additionally, the figures images and text were clear and legible.There were only a few minor corrections:
1. Line 32: define the acronym GWAS
2. Acronyms should not be at the beginning of sentences
3. Line 302: insert the word 'to' between the words 'able' and 'exert'
4. Line 345: delete an 's' in the word 'focusses'
Author Response
We thank the reviewer for the encouraging words and also highlighting some of the aspects we hoped would come across in this review paper. We truly appreciate this. We are also glad to hear that the reviewer thinks that this work is worthy of being published after making some minor corrections.
We have made changes as per the reviewer’s suggestions.
- GWAS is now replaced with genome-wide association study (GWAS)
- Changes were made to remove acronyms.
- Thank you for this grammatical correction. ‘To’ is now included in the revised manuscript.
- We believe ‘focusses’ is spelled correctly. So we have not changed this.
Author Response
We are grateful for the detailed and thorough review of our manuscript. We also appreciate the encouraging words on the detail provided in the manuscript. We have carefully gone through each point made by the reviewer, and made changes accordingly. Below are our responses which we hope are satisfactory
Overall points about the structure and organization of the review:
We understand the reviewer’s point on combining sections 2 and 3 under one section. However, the central focus of our review paper is on the brain region-metabolic organ/tissue crosstalk, that is section 3. Section 2 discusses the anatomical features of the regions, and serves as a way to introduce the different brain regions for the following jnter-organ crosstalk section. This allows us to flesh out the mechanisms (such as the ANS connections) in this section. A short introductory paragraph is introduced at the beginning of section 2 to allude to this point. Also we have changed the title to highlight the central theme of the review paper, which is the inter-organ crosstalk.
Please note that there are other reviews examining the role of hypothalamus and brainstem in energy balance (please refer to Schneeberger et al., 2014; Roh et al., 2016). However the unique aspect of the current review, in our opinion, is the section delving on the crosstalk between the different hypothalamic and brainstem regions, and the various metabolic organs and tissues. This aspect is not discussed in great detail in other reviews as per our literature review. Also by combining the two sections, would lead to a massive overhaul of the review paper. For this reason, we hope to have a section dedicated to the brain-periphery crosstalk in the final version.
Below is the point-by-point response to each comment-
- Changes were made accordingly. ‘Altering’ changed to inhibiting (line 663). ‘Augmenting’ changed to enhanced (line 669).
- GABAergic innervation of POMC neurons by AGRP neurons has been included in section 2.1.1. Please refer to lines 106-107 and lines 119-122. References 21, 24, 32, 33 and 34 were added. Figure 1 legend was updated (lines 481-482).
- Thank you for this suggestion. We have included the point about conflicting results of POMC ablation on food intake (lines 112-114).
- The neuropeptides were added to the figure. Also two text boxes were added to describe the physiological role of the hypothalamic satiety center.
- Thank you for raising this point. Points about beta-endorphin is added in the revised version (lines 122-132). References 32-40 were added.
- Thank you for bringing this to our attention. Here we did not want to mention about crosstalk, but just highlight the important role of the RhoA-ROCK axis in leptin signaling in AGRP neurons. So we removed the part on the Gq and G12/13 signaling proteins, and replaced it with a more functional role of the rho-ROCK axis in neuropeptide release. Please refer to lines 136-137.
- We made changes as per the reviewer’s suggestions. Please see point 4.
- We have included a paragraph on the HPT axis (lines 175-185). We mainly focusing on thermogenesis and the contrasting roles of AGRP and POMC on it. We have also added the roles of the melanocortin system in regulating the TSH levels. References 58-62 were also added.
- This is a very valid point. The neuronal populations/brain regions that are well-studied when it comes to sexual dimorphism are the POMC neurons/ARC. So we have included two paragraphs on this (lines 186-206). References 63-69 and 77 were added in relation to this. Here we also introduce the topic of synaptic plasticity. This has been shown to be more evident in the median eminence of female mice.
Please note that most of the mechanisms in this review were reported on male mice. This is the nature of obesity/metabolic research as C57 BL6 male mice exhibit striking metabolic alterations in response to HFD, when compared to their female counterparts (Casimiro et al., 2021). This resulted in researchers relying less on female C57 mice as an animal model of obesity. We just wanted to allude to this point, and a potential need to contextualize the gender-specific effects reported in mice studies.
- We have changed the text as per the reviewer’s suggestion
- Since we are talking about the melanocortin system, which is usually the MC4/3R and POMC neurons, we have not included AGRP neurons here.
- We have changed the title to include hypothalamus and brainstem
- Mechanisms relating to the section on clinical implications are now covered in earlier text. Relating to Naltrexone, the mechanism on MOR-mediated stimulation of POMC neurons is covered in lines 122-132. Regarding the mechanisms of the GLP1R agonist, semaglutide, was covered in earlier sections discussing the NTS and the gut-brain axis. The role of the central GLP1R in mediating satiety and improving glucose tolerance is covered in detail in section 2.2.1 and 3.4 respectively.